# EMS-Induced Mutagenesis of *Clostridium carboxidivorans* for Increased Atmospheric CO_2_ Reduction Efficiency and Solvent Production

**DOI:** 10.3390/microorganisms8081239

**Published:** 2020-08-14

**Authors:** Naoufal Lakhssassi, Azam Baharlouei, Jonas Meksem, Scott D. Hamilton-Brehm, David A. Lightfoot, Khalid Meksem, Yanna Liang

**Affiliations:** 1Department of Civil and Environmental Engineering, 1230 Lincoln Drive, Southern Illinois University Carbondale, Carbondale, IL 62901, USA; naoufal.lakhssassi@siu.edu (N.L.); baharlouei@siu.edu (A.B.); 2Department of Plant, Soil, and Agricultural Systems, Southern Illinois University, Carbondale, IL 62901, USA; ga4082@siu.edu; 3Trinity College of Arts and Sciences, Duke University, Durham, NC 27708, USA; jonas.meksem@duke.edu; 4Department of Microbiology, Southern Illinois University, Carbondale, IL 62901, USA; scott.hamilton-brehm@siu.edu; 5Department of Environmental and Sustainable Engineering, 1400 Washington Ave, State University of New York at Albany, Albany, NY 12222, USA

**Keywords:** CO_2_ reduction efficiency, EMS mutagenesis, anaerobic bacteria, global warming, Clostridium carboxidivorans, increased alcohol production

## Abstract

*Clostridium carboxidivorans* (P7) is one of the most important solvent-producing bacteria capable of fermenting syngas (CO, CO_2_, and H_2_) to produce chemical commodities when grown as an autotroph. This study aimed to develop ethyl methanesulfonate (EMS)-induced P7 mutants that were capable of growing in the presence of CO_2_ as a unique source of carbon with increased solvent formation and atmospheric CO_2_ reduction to limit global warming. Phenotypic analysis including growth and end product characterization of the P7 wild type (WT) demonstrated that this strain grew better at 25 °C than 37 °C when CO_2_ served as the only source of carbon. In the current study, 55 mutagenized *P7-EMS* mutants were developed by using 100 mM and 120 mM EMS. Interestingly, using a forward genetic approach, three out of the 55 *P7-EMS* mutants showed a significant increase in ethanol, butyrate, and butanol production. The three *P7-EMS* mutants presented on average a 4.68-fold increase in concentrations of ethanol when compared to the P7-WT. Butyric acid production from 3 *P7-EMS* mutants contained an average of a 3.85 fold increase over the levels observed in the P7-WT cultures under the same conditions (CO_2_ only). In addition, one *P7-EMS* mutant presented butanol production (0.23 ± 0.02 g/L), which was absent from the P7-WT under CO_2_ conditions. Most of the *P7-EMS* mutants showed stability of the obtained end product traits after three transfers. Most importantly, the amount of reduced atmospheric CO_2_ increased up to 8.72 times (0.21 g/Abs) for ethanol production and up to 8.73 times higher (0.16 g/Abs) for butyrate than the levels contained in the P7-WT. Additionally, to produce butanol, the *P7-EMS_III-J_* mutant presented 0.082 g/Abs of CO_2_ reduction. This study demonstrated the feasibility and effectiveness of employing EMS mutagenesis in generating solvent-producing anaerobic bacteria mutants with improved and novel product formation and increased atmospheric CO_2_ reduction efficiency.

## 1. Introduction

Since CO_2_ is the second most abundant greenhouse gas next to water vapor in the atmosphere [1], decreasing the concentration of this molecule presents a challenge when addressing the effects of global warming. Capturing and sequestrating CO_2_ is an approach to mitigating CO_2_ concentrations in the air [2]. However, converting CO_2_ to commodities may be a better solution given the abundance of CO_2_ emitted from numerous sources [3].

CO_2_ is a highly oxidized C-1 compound that can be reduced chemically and biologically [1]. Breaking down CO_2_ to reduce its concentration in the air requires catalysts that can be expensive, easy to be poisoned, and may necessitate high temperatures and/or pressure [1]. Capturing and sequestering CO_2_, however, utilizes microorganisms that can self-regenerate and perform well at mild environmental conditions [1]. Among countless microbes that can utilize CO_2_ as a carbon source, *Clostridium* strains have attracted extensive attention in recent years. This stems from their capability in reducing CO_2_ to acids and alcohols through the Wood-Ljungdahl pathway. Increasing demand for renewable biofuel production has generated a particular interest in dense biofuel production. To date, no industrially relevant *Clostridium* has been used, due to the challenging growth conditions and difficulties related to handling these microorganisms.

Among various *Clostridium* strains, *Clostridium carboxidivorans* P7 (ATCC BAA-624) is unique. This solvent-producing bacterium was isolated from sediment from an agricultural settling lagoon after enrichment with CO as the substrate [4]. This bacterium can grow chemoorganotrophically with simple sugars [4]. Interestingly, it is one of three obligate anaerobe microbial catalysts capable of fermenting synthesis gas (CO, CO_2_, and H_2_) to produce liquid biofuels like ethanol, butanol, and hexanol when grown autotrophically [5]. This acetogenic *Clostridium* contains the gene of selenocysteine-containing formate dehydrogenase, and therefore, can fix carbon via the Wood-Ljungdahl pathway to reduce CO_2_ to formate [6]. Furthermore, the P7 strain encodes NADPH-dependent butanol dehydrogenase that converts acetyl-CoA into butanol [7]. It is well known that industrial strains like *Clostridium acetobutylicum* also encode an NADPH-dependent butanol dehydrogenase but cannot convert CO_2_ or CO to acetyl-CoA. On the other hand, *Clostridium ljungdahlii* can fix both CO and CO_2_, but lacks butanol dehydrogenase and cannot convert acetyl-CoA into butanol [8]. Therefore, the P7 strain includes beneficial properties of both *C. acetobutylicum* and *C. ljungdahlii* strains. 

Two major limitations have delayed the commercialization of this strain [9]. It has been found that rapid accumulation of organic acids like acetic acid during exponential growth results in low alcohol production and substrate consumption due to acid crash [9]. Additionally, the titers of ethanol, butanol, and hexanol are generally low and sometimes undetectable. Notably, the release of the complete genome of *C. carboxidivorans* (genome #ADEK00000000) [8] encouraged the scientific community to start developing transformation protocols in order to engineer this P7 strain [8,10,11,12,13].

Molecular cloning techniques have been widely used to overexpress or knockout target genes within pathways in order to study gene function in bacteria, yeasts, animals, and plants [11]. Most of these techniques are labor intensive and are based on selection methods. There are two main challenges that make engineering *C. carboxidivorans* difficult. First, being an obligate anaerobic bacterium, growth of the P7 strain on solid media is poor, rendering propagation difficult. Secondly, the P7 strain goes through sporulation in its life cycle, this may lead to a loss of transforming vectors carrying genes of interest. Recently, CRISPR/Cas9-based genome editing was tested on *C. ljungdahlii* [14]. This technique was proven to be feasible, but all mutants generated had lower growth rates and lower titers of acids and alcohols than the wild type strain [14].

To date, the genome editing approach [14] has not yet been reported for the P7 strain. To avoid similar issues that we may encounter if the genome editing technique is adopted, a chemical mutagen, ethyl methanesulfonate (EMS) was used in this study to generate *P7-EMS* mutants for the purpose of obtaining superior mutants with desired properties. EMS has been widely used in plants and other microorganisms [15,16,17,18,19]. In addition, considering the fact that P7 strain has been primarily studied on syngas (CO+ H_2_ or CO +CO_2_ +H_2_) fermentation [5,7,9,12,20,21,22,23,24], the current study focused on CO_2_ reduction in the absence of carbon monoxide (CO). This will allow for increasing the reduction of atmospheric CO_2_ efficiency, reducing the effects of global warming together with producing increased alcohols beneficial for the biofuel industry and environment.

## 2. Materials and Methods

### 2.1. Bacterial Strain, Media, and Culture Conditions

*Clostridium carboxidivorans* P7 (ATCC_®_ BAA-624™) was obtained from the American Type Culture Collection (ATCC) [25]. During the last decade, several growth mediums have been reported to grow the P7-WT [4,24,26]. According to ATCC’s instruction, the freeze-dried powder was regenerated in the Wilkins Chalgren (W-C) Anaerobic Medium. After cells grew to a reasonable density, they were transferred to a ATCC 1754-A PETC medium with fructose at 5 g/L, or into a modified medium with 3.6 g/L HEPES buffer without fructose, yeast extract, or calcium chloride, and containing 90% less minerals and vitamins than the 1754-A (Table 1). The latter, which is the optimized medium in the current study, was referred to as 1754-B medium (Table 1). Serum bottles (100-mL, Wheaton) containing the 1754-A PETC medium were used to grow the P7-WT and the *P7-EMS* mutants. Nitrogen was used to purge the headspace gas. Unlike conventional syngas conditions (CO+ H_2_, or CO +CO_2_ +H_2_) that were previously reported to grow the P7-WT, the isolated P7-EMS mutants and the P7-WT that were grown on the 1754-B medium were purged with CO_2_/H_2_ (20:80, *v/v*) only to fill the headspace, but without the presence of carbon monoxide (CO) since the current study aims to study the impact of the produced *P7-EMS* in reducing atmospheric CO_2_ and converting it to alcohols.

To prepare for the mutagenesis study, different recipes for making agar plates were also tested. Those included ATCC 260 Tryptic Soy Broth (TSB) with 5% Sheep Blood (defibrinated) (www.atcc.org), the W-C medium, the 1754-A PETC, and the 1754-B. For each recipe, agar was added at 15 g/L. In addition, big (100-mL) and small (20-mL) serum bottles were used to optimize the bacteria growth. The rationale for studying different bottle sizes was that big bottles would occupy more space in the anaerobic glove box, facilitating more medium, and are less cost effective when compared to smaller ones. This is especially crucial since a great number of mutants would be generated and cultivated in a liquid medium. Thus, for this purpose, 100 mL and 20 mL serum bottles were used. To the former, 25 mL of either the W-C or 1754-B medium was added, and to the latter, 10 mL of each was used. All serum bottles were inoculated with the P7 strain grown in either the W-C or 1754-B medium (Appendix A). Each condition was tested with three replicates. Nitrogen or CO_2_/H_2_ (20/80) was used to purge the headspace of serum bottles containing the W-C or 1754-B medium, respectively. A total of 24 bottles (two bottle sizes × two medium recipes × two temperatures × three replicates) were maintained at either 25 °C or 37 °C. These cultures were monitored for cell growth by measuring optical density (OD) at 600 nm and cell counting by using a hemocytometer installed on a light microscope. In addition, samples (1 mL) were withdrawn at different time points for product analysis by HPLC as detailed below.

### 2.2. Ethyl Methanesulfonate (EMS) Mutagenesis of the P7 Population

The *Clostridium carboxidivorans* P7 wild type strain was used to develop a large EMS mutagenized population. To develop the *P7-EMS* mutants, serial EMS concentrations of 0.2% (20 mM), 0.4% (40 mM), 0.6% (60 mM), 0.8% (80 mM), 1.0% (100 mM), 1.2% (120 mM), 1.4% (140 mM), and 1.6% (160 mM), 1.8% (180 mM), and 2.0% (200 mM) (*w/v*) were used to mutagenize a P7-WT culture in its exponential phase of growth. Based on the number of colonies observed on TSB plates with 5% (*w/v*) sheep blood, EMS at 1.0% (*w/v*) and 1.2% (*w/v*) were used in later mutation studies. In fact, EMS concentrations at 140mM and above were lethal for the P7 strain. Moreover, mutational rates showed that the LD_50_ corresponds to 74.89 mM of EMS treatment.

Briefly, the P7 strain was exposed to EMS at a target concentration for 6 h followed by mutant growth and recovery on the TSB plates in a CO_2_:H_2_ (20:80) atmosphere at 25 °C. After one week, 40 colonies randomly selected from those mutated by 1.0% (*w/v*) EMS were picked from the plates inside an anaerobic chamber containing an 8% Hydrogen, 20% CO_2_, and 72% (*v/v*) N_2_ balance. The selected colonies were then transferred to the optimized 1754-B medium in serum bottles. The headspace was purged with CO_2_:H_2_ (20:80, *v/v*). For those exposed to EMS at 1.2%, 15 mutants were randomly picked and transferred to the same 1754-B medium. Growth of the mutants in the liquid medium was monitored over time. Samples (1-mL) at different time points were withdrawn for analyzing products and the headspace was purged with CO_2_/H_2_ (*v/v*) periodically. In addition, frozen stocks in 15% (*v/v*) glycerol were made for all 55 mutants. All of the frozen stocks were regenerated in the 1754-B medium to confirm the product formation and stability after the third transfer.

### 2.3. End Product Analysis

End products formed from CO_2_ reduction were quantified by HPLC (Shimadzu Scientific Instrument, Inc, Columbia, MD, USA). This HPLC had a refractive index detector and was equipped with an Aminex HPX-87 column (5 µm, 30 cm × 4.6 mm, Bio-Rad, Hercules, CA, USA) maintained at 40 °C. Sulfuric acid at 0.005 M was used as the mobile phase with a flow rate of 0.6 mL/min. The sample injection volume was 20 µL. Calibration curves for possible products, such as formic acid, acetic acid, butyric acid, ethanol, and butanol were established using standard compounds.

### 2.4. CO_2_ Reduction and Statistical Analysis

After measuring the concentrations of the different end products obtained by HPLC (based on the equations shown in Appendix A), each end product (g/L) was normalized based on the strain growth (OD) and absorbance (g/L.Abs) as shown below:
(1)$$\mathrm{Normalized}\;\mathrm{End}\;\mathrm{Products}\;\left(\mathrm{NEP}\right)\;\left(\frac{g}{L}\right)x\;\mathrm{Abs}=\frac{\mathrm{End}\;\mathrm{product}\;\left(\frac{g}{L}\right)}{\mathrm{OD}\;\left(\mathrm{Abs}\right)}$$

Next, CO_2_ reduction was calculated based on the following equation:
(2)$${\mathrm{CO}}_{2}\;\mathrm{reduction}\;\left(\frac{g}{\mathrm{Abs}}\right)=\left[\left(\frac{\mathrm{NEP}\;\left(\frac{g}{L}\right)x\;\mathrm{Abs}}{\mathrm{Product}\;M.W\;\left(\frac{g}{\mathrm{mol}}\right))}\right)\;\times \;\mathrm{Volume}\;\mathrm{of}\;\mathrm{growth}\;\mathrm{medium}\;\left(L\right)\;\times \;{\mathrm{CO}}_{2}\;M.W\;\left(\frac{g}{\mathrm{mol}}\right)\right]\times \;\mathrm{Nbr}.\;\mathrm{mol}.\;\mathrm{of}\;C$$

(V) is the volume of the PETC 1754-B growth media used (0.025L). (P.M.W) is the product molecular weight (46.07 g/mol, 88.11 g/mol, and 74.12 g/mol for ethanol, butyrate, and butanol respectively), (CO_2_ M.W) CO_2_ molecular weight is 44.01 g/mol. (N.M.C) is the number of molecules of carbons in each alcohol product (2 molecules of carbon for ethanol, and 4 molecules for butyrate and butanol). (Abs) corresponds to the optical density (OD) of each strain.

Statistical analysis was performed using Student’s *t*-test for comparisons of means, using the JMP Pro V14 software (SAS Institute Inc., Cary, NC, USA).

## 3. Results

### 3.1. Cell Growth and Product Formation of the Wild Type P7 Strain

As shown in Appendix A, the wild type P7 strain grew well in the W-C rich-medium containing glucose, tryptone, gelatin peptone, yeast extract, and sodium pyruvate. At 37 °C, the cells reached the end of the exponential growth phase by day 5. At 25 °C, the cell density reached the highest level of growth at day 4, showing 1.13 times more cell density when compared to cells grown under CO_2_ conditions at 37 °C. When a defined medium (1754-B) (Table 1) without the presence of any organic components was used, the growth of the P7 strain was slow, but the P7-WT growth was better at 25 °C than at 37 °C, showing 3.4 times more cell density when compared to cells grown at 37 °C (Figure 1). The number of cells grown at 25 °C were 5.6 times more when compared to cells grown under CO_2_ conditions at 37 °C in the defined medium (Figure 1).

When cells grew on the 1754-B medium at 25 °C under CO_2_ conditions only (absence of CO), three products, including formic acid, acetic acid, and ethanol and a trace amount of butyric acid, were produced (Figure 2). Among the four products, acetic acid (1.05 g/L) was the major product (19.04 g/L.Abs) after 26 days of growth. The highest ethanol production of up to 0.22 g/L (0.4 g/L.Abs) was observed at day 4. When P7-WT was grown in the 1754-B medium, butyrate acid was not detected in any of the cultures until day 26 at less than 0.14 g/L (0.5 g/L.Abs) (Figure 2). However, no butanol production was observed when the P7-WT is grown under CO_2_ conditions only.

The total concentrations of all products were smaller when using small (20-mL) serum bottles than when using big (100-mL) serum bottles (Appendix A), or when the P7 is grown under syngas conditions (CO+ H_2_ or CO +CO_2_ +H_2_) [22]. For cultures grown at 25 °C, slightly more end products produced by the P7-WT were detected when compared to cultures grown at 37 °C in the 1754-B defined medium. Therefore, for later experiments, all mutants were cultivated at 25 °C in the defined 1754-B medium in 100 mL serum bottle (BB 25 °C).

### 3.2. Development of an EMS Mutagenized Population of C. carboxidivorans P7

In order to identify the best EMS mutational rate to mutagenize P7 growing cells in the defined medium 1754-B, different increasing EMS concentrations have been tested. The induced EMS mutagen negatively impacted the number of *P7-EMS* mutagenized colonies that decreased proportionally while increasing the EMS concentrations from 20mM until 140mM. Among the EMS concentrations tested from 0.2 to 2.0% (*w/v*), an increase in EMS concentrations had a negative effect on P7 growth with concentrations above 1.4% being lethal (Figure 3). Additionally, the obtained LD_50_ corresponded to 74.89mM EMS concentration. Therefore, EMS concentrations at 1.0 and 1.2% (*w/v*) were used to develop a large *P7-EMS* mutant population. The use of the two highest concentrations of EMS that allowed *P7-EMS* colonies to survive warrants a maximum mutation density (saturation) and maximum SNPs that were introduced randomly by EMS mutagenesis, and therefore, a better engineering of the P7 to produce ethanol, butyrate, and butanol via the Wood-Ljungdahl pathway. Based upon the results from the analysis of samples collected at different time points for each mutant, out of 40 *P7-EMS* mutants derived from the EMS treatment at 1.0% and 15 *P7-EMS* mutants derived from the treatment at 1.2%, the majority of the developed *P7-EMS* mutants had similar end products as observed in the P7-WT.

### 3.3. P7-EMS Mutagenized Mutants Showed Significant Increased Levels of Alcohol Production

Interestingly, three out of 55 *P7*-EMS mutants had different products and product profiles compared to the P7-WT (Figure 4). In particular, *P7_III-J_, P7_III-R_*, and *P7_III-P_* EMS mutants presented increased ethanol production of up to 0.87 g/L, 1.64 g/L, and 1.79 g/L respectively, compared to a production of 0.22 g/L in the P7-WT (Figure 4). Furthermore, the *P7_III-J_, P7_III-R_,* and *P7_III-P_ EMS* mutants presented increased butyrate production of up to 0.7 g/L, 0.49 g/L, and 0.28 g/L, respectively, compared to 0.14 g/L butyrate production in the P7-WT. Surprisingly, butanol was produced in the *P7_III-J_* EMS mutant (up to 0.25 g/L), which was absent from the wild type when grown under CO_2_ conditions (Figure 4).

Moreover, for each of these four mutants, glycerol-frozen stocks were made. Cells from the frozen stocks were revitalized in the 1754-B defined medium again and the products were tracked. Interestingly, formic acid production was lower in all three analyzed *P7-EMS* mutants when compared to the P7-WT (Figure 5).

In order to study the stability of the developed mutants, we further evaluated the three *P7-EMS* mutants and tested their performances after three transfers from a frozen stock onto the 1754-B defined medium with three replicates. As expected, the results obtained confirmed that random mutations introduced by the EMS mutagenesis were stable in the subsequent generations (Figure 6). In fact, the three mutants presented on average a significant (*p* < 0.5) stable increase in ethanol production by 3.5 (*P7_III-J_*), 4.13 (*P7_III-P_*), and 4.68 (*P7_III-R_*) times the levels contained in the P7-WT (Table 2). The three mutants maintained significantly (*p* < 0.5) increased butyrate production at 1.92 (*P7_III-P_*), 3.21 (*P7_III-R_*), and 3.85 (*P7_III-J_*) times more when compared to the P7 wild type. Ultimately, the butanol production was maintained in the *P7_III-J_* mutant after three transfers.

### 3.4. P7-EMS Mutants Showed Increased Levels of Atmospheric CO_2_ Reduction

To gain more insight into the link between the reduced CO_2_ and the observed end product differences between the P7-WT and the *P7-EMS* mutants, the amount of reduced CO_2_ in all lines has been calculated. As shown in Table 2, the amount of reduced CO_2_ increased in all three identified *P7-EMS* mutants up to 0.21 g/Abs, which is up to 8.72 times more when compared to 0.02 g/Abs of CO_2_ reduction that was observed in the P7-WT for ethanol production. Similarly, for butyrate production, the amount of reduced CO_2_ increased in all three identified *P7-EMS* mutants up to 0.16 g/Abs, which is up to 8.73 times more when compared to the ~0.02 g/Abs CO_2_ reduction that was observed in the P7-WT (Table 2). For butanol production, an additional 0.082 g/Abs of CO_2_ reduction was observed in the *P7_III-J_* EMS mutant (Table 2). Notably, the obtained data from the current study demonstrates that the generated *P7-EMS* mutants improve their atmospheric CO_2_ utilization efficiency, positively impacting the alcohol solvent formation.

## 4. Discussion

An earlier study introduced the response of *Escherichia coli* to EMS that influenced their growth phase and repair ability on survival and mutagenesis [17]. Later, EMS-induced mutagenesis in the aerobic bacteria model *E. coli*, was shown to induce predominantly GC-->AT transitions and AT-->TA transversions [27]. Furthermore, large-scale mutational analysis of EMS-induced mutation in the *lacI* gene of *E. coli* was achieved [28]. In anaerobic bacteria, little effort has been made in developing new transformation protocols. Recently, a new electrotransformation protocol among the genus of Clostridium sp. (*C. pasteurianum*) has been reported [11]. However, no study has reported the use of EMS as a mutagen in *C. carboxydivorans* in order to improve its end product profiles.

As discussed above, the P7 strain has been mainly tested on syngas fermentation. Compared with the sugar platform where lignocelluosic biomass is pretreated and hydrolyzed to release sugars for microbial fermentation, the syngas platform employs less steps and is able to accommodate different biomass waste materials [29]. However, besides costs associated with high temperatures needed for gasification, syngas fermentation releases a gas stream containing a higher CO_2_ concentration than the input syngas. The CO_2_ concentration increase was due to the utilization of CO and/or H_2_ and releasing CO_2_ through respiration during microbial fermentation [7]. Thus, from the perspective of CO_2_ reduction, the whole biomass-to-syngas-to-products process needs to be examined carefully. Nevertheless, in this study, reducing CO_2_ to commodities was the main focus. 

Comparable to what was reported previously [9,24], the P7 strain did grow in a chemically defined medium without any organic carbon and nitrogen sources. Even though the cell growth rate and production titers are lower than those from the W-C medium, the inexpensive and lean medium should benefit future efforts toward commercialization, especially for CO_2_ reduction, as this molecule serves as the only carbon source. Compared to syngas fermentation, the P7 wild type yield of products observed in this study from CO_2_ only was also lower than 1.48 g /L for ethanol and 1.07 g/L for butanol reported by Ramio-Pujol et al. [9] and 1.0 g/L for butanol, up to 1.0 g/L for hexanol, and over 3.0 g/L for ethanol demonstrated by Phillips et al. [24]. The differences are most likely due to the presence of CO in the syngas, which is more easily utilized as a substrate than CO_2_, as it has been demonstrated.

All of the frozen stocks from the three P7-EMS mutants that were regenerated in the 1754-B medium showed the recovery and stability of their corresponding obtained end products after three transfers. Moreover, it’s important to mention that we have developed within the original 55 P7-EMS mutants another mutant P7_I-A_ that showed a dramatic increase in alcohol production up than 8.77g/L. However, the isolated P7_I-A_ mutant was not able to re-grow at highest concentrations of alcohol resulting in a lethal effect on the P7 _I-A_ cells. Therefore, highest alcohol production by the P7-EMS mutants (up to 8.77g/L) may impact and influence negatively the efficiency of the mutants. This physicochemical factor should be considered when employing this forward genetics approach to developing anaerobic EMS mutants. 

The mutagenesis of the P7 strain was demonstrated to be successful by increasing ethanol production up to 1.79 g/L on CO_2_ only, which is higher to what was reported previously when the P7-WT is grown in the presence of CO_2_ plus CO [9]. The growth of these mutants under syngas conditions (CO+ H_2_, or CO +CO_2_ +H_2_) is expected to provide the highest alcohol production, as it has been shown that increasing CO content during syngas fermentation has a positive effect on the P7-WT alcohol production [9,24]. The systematic mutagenesis employed in the current study provided several mutants with improved ethanol, butyric acid, and butanol production. Compared to the P7-WT that did not produce butanol and contained lower levels of ethanol and small trace amounts of butyric acid, the isolated EMS mutants yielded better in their end products. However, the product concentrations were still relatively low to warrant their use at industrial scales when grown to reduce atmospheric CO_2_, unless they are grown under syngas conditions (CO+ H_2_, or CO +CO_2_ +H_2_), where we expect higher production. 

Although the current study was able to develop EMS anaerobic mutants from the P7-WT to boost their alcohol end product and increase atmospheric CO_2_ reduction using CO_2_ as the only source of carbon, genotyping of the P7-EMS mutants in key enzymes will have a highest impact to further engineer the anaerobic strains. To further improve the formation of end products, the desired mutants obtained from this study may need to be subject to other techniques, such as TILLING to identify mutations impacting key enzymes in the acetyl-CoA pathway, molecular cloning, targeted gene overexpression, and gene silencing in order to improve product titers. Most likely, EMS mutations that were randomly introduced in the *P7-EMS* mutants genome may occur within the ORFs of known genes involved in the acetyl-CoA pathway including the alcohol/aldehyde dehydrogenase (AAD) and alcohol dehydrogenase (ADH) pathway that convert acetyl-CoA into ethanol and butyryl-CoA into butanol, and in the phosphotransbutyrylase enzyme (PTB) that converts butyryl-CoA into butyrate, or in the pyruvate:ferredoxin oxidoreductase (PFOR) that converts acetyl-CoA into 2,3-butanediol. [30]. Moreover, formic acid (HCOOH) production was lower in all three *P7-EMS* mutants analyzed when compared to the P7-WT. The formic acid is the first metabolite synthesized after reduction of the atmospheric CO_2_ at the “eastern” methyl branch [30]. Therefore, another explanation could be related to the redirection of the carbon flux from the “western” carbonyl branch and an overuse of the “eastern” methyl branch, due to the introduction of deleterious mutations on the CO dehydrogenase (CODH) that converts CO_2_ to CO. 

In light of the results obtained, this study provided a solid foundation as to what medium and bottle size to use, what headspace gas to choose, and how to develop, isolate, and select the P7 anaerobic EMS mutant bacteria. Interestingly, EMS mutagenesis increased the CO_2_ reduction of all mutants up to 0.21 g/Abs CO_2_ to produce up to 4.58 g/L Abs ethanol (8.72 times more CO_2_ reduction than the P7-WT), and up to 0.16 g/Abs CO_2_ reduction to produce 3.33 g/L Abs butyrate, were observed in the *P7_III-J_* EMS mutant when compared to 0.025 and 0.019 g/Abs CO_2_ that was reduced in the P7-WT to produce ethanol and butyrate, respectively. This represents an 8.73-fold increase in CO_2_ reduction on the *P7_III-J_* EMS mutant when compared to the P7-WT.

## 5. Conclusions

The current study demonstrates the effectiveness of employing EMS mutagenesis to reduce increasing amounts of undesired atmospheric CO_2_, having positive effects on increased production levels of alcohols by activating the hydrocarbon pathway and decreasing the acidic pathway. Results obtained from the current study represent a breakthrough discovery in how we can utilize a microscopic anaerobic bacterium like P7 to reduce increased quantities of atmospheric CO_2_ by improving their CO_2_ utilization efficiency, and thus, limit global warming in our planet.

## Figures and Tables

**Figure 1 microorganisms-08-01239-f001:**
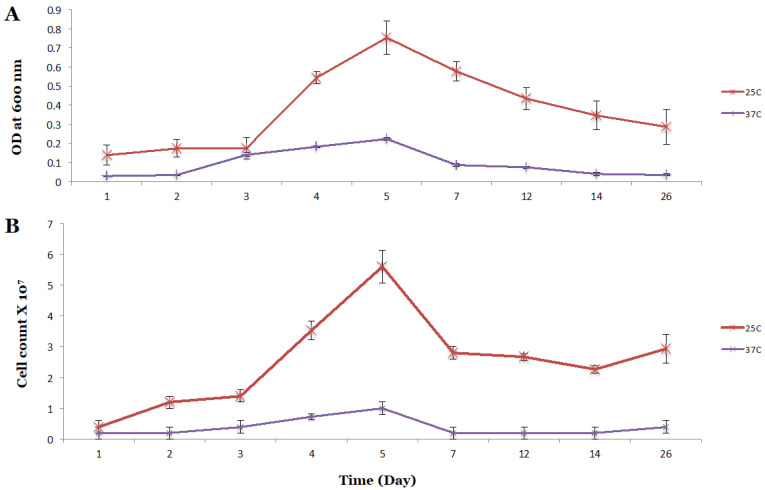
Growth comparison of *Clostridium carboxidivorans* P7 wild type employing two different temperatures 25 °C and 37 °C in the optimized 1754-B medium. (**A**) Optical density, (**B**) Cell count.

**Figure 2 microorganisms-08-01239-f002:**
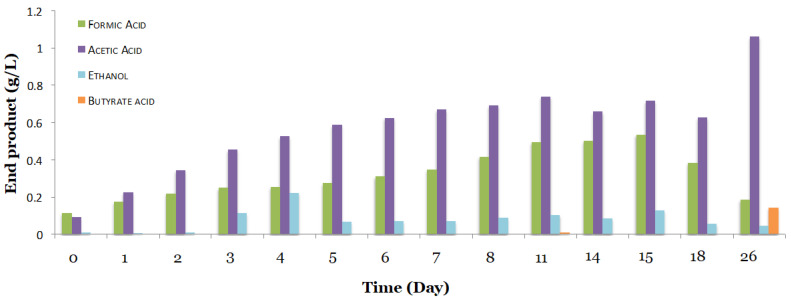
End product identified for *Clostridium carboxidivorans* wild type P7 in the 1754-B modified medium. Butyrate production was detected, but not until after day 26.

**Figure 3 microorganisms-08-01239-f003:**
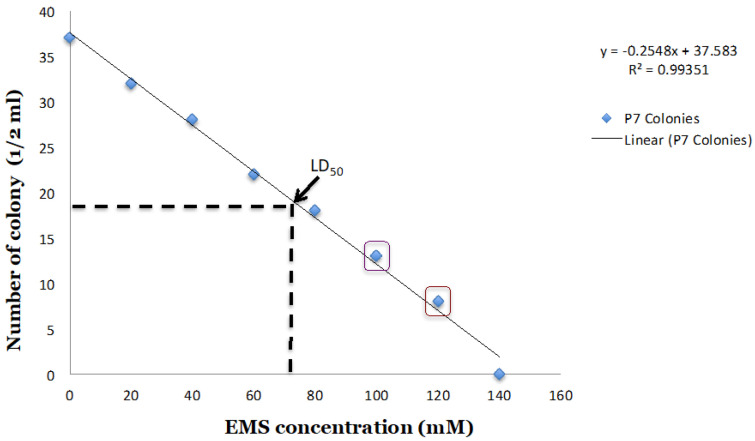
Correlation between the number of colonies and ethyl methanesulfonate (EMS) concentrations. Eight ethyl methanesulfonate (EMS) concentrations have been used to mutagenize the *Clostridium carboxidivorans* P7 wild type culture. Concentrations above 140mM were lethal for the P7 strain. Mutational rates show an LD_50_ of 74.89mM of EMS treatment. 40 (purple box) and 15 (brown box) EMS mutants were randomly isolated from 100 mM and 120mM EMS treatment, respectively.

**Figure 4 microorganisms-08-01239-f004:**
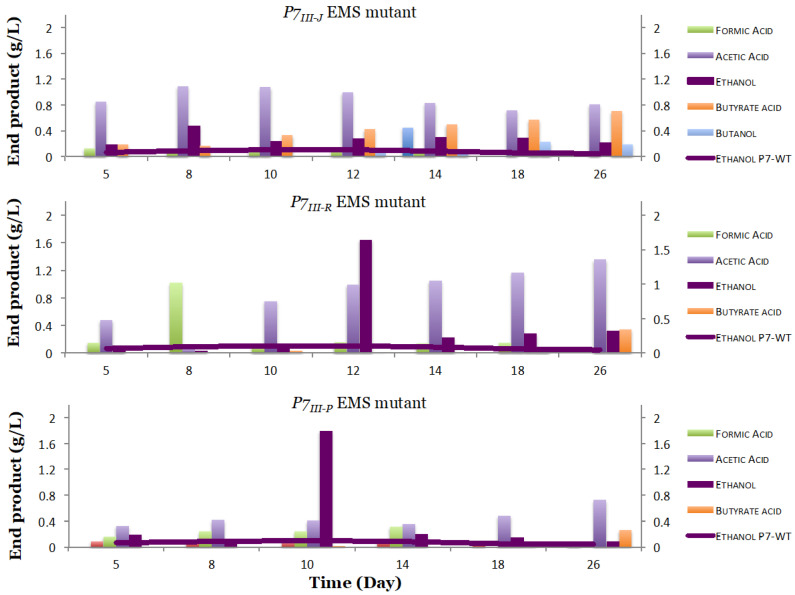
End products identified for the mutagenized *Clostridium carboxidivorans P7-EMS* mutants. Three out of 55 *P7-EMS* mutants presented different products and product profiles when compared to the wild type P7. The three mutants (*P7_III-J_*, *P7_III-R_*, and *P7_III-P_*) showed increased ethanol production, in addition, butyrate acid production was observed in all three mutants, but not in the P7-WT when CO_2_ is provided as the only source of carbon. The purple line represents ethanol production of the P7 wild type.

**Figure 5 microorganisms-08-01239-f005:**
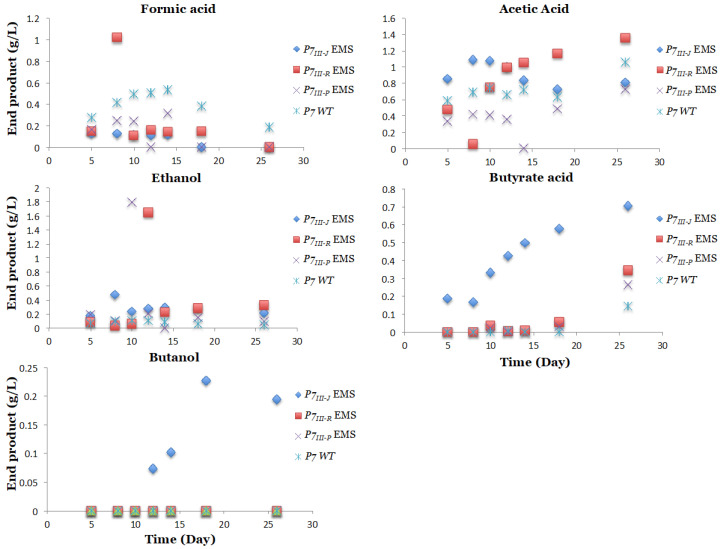
Acid and Alcohol end product (g/L) comparison between the three *P7-EMS* and the P7-WT. The three *P7-EMS* mutants (*P7_III-J_*, *P7_III-R_*, and *P7_III-P_*) showed decreased formic acid production during four weeks of growth, and increased ethanol production, in addition, butyrate acid production was observed in all three mutants, but not in the P7-WT when CO_2_ is provided as the only source of carbon.

**Figure 6 microorganisms-08-01239-f006:**
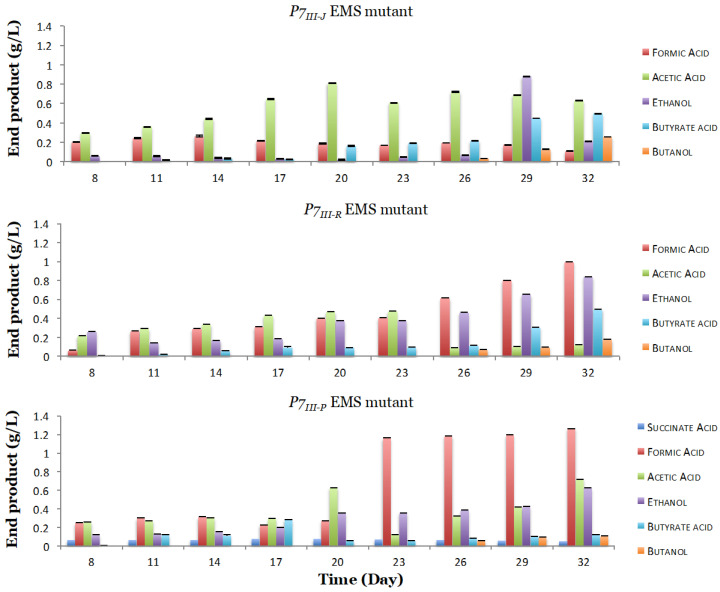
End products identified for the three *P7-EMS* mutants on CO_2_ after three transfers. All three *P7-EMS* mutants presented stable and increased ethanol, butyrate acid, and butanol production when compared to the P7 wild type.

**Table 1 microorganisms-08-01239-t001:** Composition of the optimized 1754-B medium used in this study. The original PETC medium (1754-A) was used with some modifications as shown below (Gray).

Component	Formula	1754-A (Original PETC Medium)	1754-B (Modified PETC Medium)
		**Volume (0.5 L)**	**Volume (0.5 L)**
***Resazurin***			
Resazurin	C_12_H_6_NO_4_Na	200 µL	200 µL
***Minerals***			
Ammonium chloride	NH_4_Cl	500 mg	500 mg
Magnesium sulfate	MgSO_4_·7H_2_O	100 mg	100 mg
Potassium chloride	KCl	50 mg	50 mg
Potassium phosphate	KH_2_PO_4_	50 mg	50 mg
Sodium chloride	NaCl	400 mg	400 mg
Calcium chloride	CaCl_2_·2H_2_O	20 mg	Not added
***Buffers***			
HEPES	C_8_H_18_N_2_O_4_S	Undefined	1.8 g
***Carbon source***			
Yeast Extract		500 mg	Not added
Fructose	C_6_H12O_6_	2.5 g	Not added
Sodium Bicarbonate	NaHCO_3_	1 g	1 g
***Minerals and Vitamins***			
ATCC Minerals		10 mL	1 mL
ATCC Vitamins		10 mL	1 mL
***Cysteine/Sulfide***			
L-Cysteine	C_3_H_7_NO_2_S	2 g	2 g
Sodium sulfide	Na_2_S·9H_2_O	2 g	2 g

**Table 2 microorganisms-08-01239-t002:** End products detected for *Clostridium carboxidivorans* P7 wild type and the novel mutagenized *P7-EMS* mutants.

End Product	P7 Strains	Average (g/L)	End Product Content Increase	Maximum Production (g/L)	End Product Converted on (g/L Abs)	CO_2_ Reduction (g/Abs)	CO_2_ Reduction Increase
Ethanol	III-J	0.77 ± 0.20 *	3.50X	0.87	4.58	0.218	8.72 X
	III-R	1.03 ± 0.40 *	4.68X	1.64	4.21	0.200	8 X
	III-P	0.91 ± 0.58 *	4.13X	1.79	3.38	0.161	6.44 X
	**P7-WT**	**0.21 ± 0.01**	_	**0.22**	**0.40**	**0.025**	_
Butyrate	III-J	0.54 ± 0.10**	3.85X	0.7	3.33	0.166	8.73 X
	III-R	0.45 ± 0.07**	3.21X	0.49	2.88	0.143	7.52 X
	III-P	0.27 ± 0.01 *	1.92X	0.28	1.12	0.055	2.89 X
	**P7-WT**	**0.13 ± 0.01**	_	**0.14**	**0.50**	**0.019**	_
Butanol	III-J	0.23 ± 0.02	^x^	0.25	1.38	0.082	^x^
	**P7-WT**	**ND**	_	**ND**	**NA**	**NA**	_

^x^ No butanol production was observed in the P7-WT when grown under CO_2_ conditions. However, a butanol production was detected on the *P7_III-J_* EMS mutant. The amount of CO_2_ reduced by the four identified *P7-EMS* mutants when compared to the P7 wild type is shown. CO_2_ reduction increased for the four *P7-EMS* mutants when compared to the P7 wild type. Statistical analysis was obtained as determined by Student’s *t*-test (** *p* < 0.001, * *p* < 0.05).

## Data Availability

The developed P7-EMS mutagenized mutants, are the property of the Southern Illinois University Carbondale (SIUC). Access to the strains and mutants is subject to a Material Transfer Agreement (MTA).

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
