# Peer review of "EMS-Induced Mutagenesis of Clostridium carboxidivorans for Increased Atmospheric CO2 Reduction Efficiency and Solvent Production"

_microorganisms, 2020, doi:10.3390/microorganisms8081239_

Round 1

Reviewer 1 Report

Naoufal Lakhssassi et al's manuscript applied ethylmethanesulfonate (EMS)-induced Clostridium carboxidivorans (P7) mutants to utilize CO2 as a unique source of carbon with the increased solvent formation and atmospheric CO2 reduction to limit global warming. The paper is well written. However, the experiment design of the paper is insufficient. The authors need to supply the following information before the paper can be accepted:

(1) The description of the genome information of the mutants is lacking. I understand that genomic sequences may be confidential for the authors but it is sort of obligate to open the information to the public. Otherwise, it is difficult to assess the righteousness of your results.

(2) A recovery/stability test is missing. Can the super characters of the mutants last long? Are there any physicochemical factors to influence the efficiency of the mutants?

Author Response

Answer 1:

We would like to thank reviewer 1 for his/her pertinent comments, the purpose of this manuscript is to communicate the feasibility of developing EMS anaerobic mutants from the P7-WT to boost alcohol production and reduce the atmospheric CO2 by using CO2 as only source of carbon, which has been successfully implemented and recognized by reviewer 2.

Maybe it was not clear from the text that these mutants have not been sequenced yet. We have edited the corresponding section to make it clear to the reader (Please see lines 338-341).

We fully agree with the reviewer that sequencing the mutants will add more value to our findings, once funding’s are available, we will perform whole genome sequencing and analyze the data to report in the next follow-up publication the genomics/pathway analysis results. As we mention in the discussion part of the manuscript, “we are planning to genotype the developed mutants at the targeted genes of the acetyl-CoA pathway as it will be done when funding’s are available”.

This project has been initially funded by the U.S. department of energy (D.O.E) to develop an anaerobic strain (C. carboxydivorans) with increased alcohol production based on phenotypic analysis. Indeed, it’s the main message communicated through this manuscript. This is a unique study where we did not use the syngas including the carbon monoxide which where reported earlier as a major source of carbon to grow the P7 strain. This study is beneficial for both, the environment to limit/reduce atmospheric CO2 concentrations impacting global warming, in addition to industrial energy efficiency companies for biofuel production.

As much as we agree with the comments of reviewer 1, we would like to inform the reviewer that to assess the righteousness of our results, we will be happy to give access to the mutants strains for evaluations under standard university MTA, we are also making the mutants available to the scientific community by the same standard university MTA (Lines 389-391).

Reviewer 2 Report

To authors: This is a nicely organized basic research which uses C. carboxidivorans to fix carbon dioxide. 

There is one correction needed and it is in line 293. The species should be given as lower case. 

Author Response

Answer: The name of species has been corrected.

The authors would like to thank the reviewer 2 for his/her comments.

Round 2

Reviewer 1 Report

The authors answered my first question but the second one. As the authors replied, the main purpose of this manuscript is to report the mutant based on the phenotype. So, the stability test may not be a concern. I have no further comments.